# Estimation of Excitation Current of a Synchronous Machine Using Machine Learning Methods

Matko Glučina [1], Nikola Anđelić [2], Ivan Lorencin [2,*] and Zlatan Car [2]

1  University of Rijeka, Trg Braće Mažuranića 10, 51000 Rijeka, Croatia
2  Faculty of Engineering, University of Rijeka, Vukovarska 58, 51000 Rijeka, Croatia
*  Correspondence: ilorencin@riteh.hr; Tel.: +385-51-505715

**Abstract:** A synchronous machine is an electro-mechanical converter consisting of a stator and a rotor. The stator is the stationary part of a synchronous machine that is made of phase-shifted armature windings in which voltage is generated and the rotor is the rotating part made using permanent magnets or electromagnets. The excitation current is a significant parameter of the synchronous machine, and it is of immense importance to continuously monitor possible value changes to ensure the smooth and high-quality operation of the synchronous machine itself. The purpose of this paper is to estimate the excitation current on a publicly available dataset, using the following input parameters: $I_y$: load current; PF: power factor; e: power factor error; and $d_f$: changing of excitation current of synchronous machine, using artificial intelligence algorithms. The algorithms used in this research were: k-nearest neighbors, linear, random forest, ridge, stochastic gradient descent, support vector regressor, multi-layer perceptron, and extreme gradient boost regressor, where the worst result was elasticnet, with $R^2 = -0.0001$, MSE = 0.0297, and MAPE = 0.1442; the best results were provided by extreme boosting regressor, with $\overline{R^2} = 0.9963$, $\overline{MSE} = 0.0001$, and $\overline{MAPE} = 0.0057$, respectively.

**Keywords:** artificial intelligence algorithms; excitation current; regression algorithms; synchronous machine

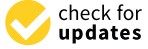



## 1. Introduction

Electrical machines are electromechanical devices that can convert electrical energy into mechanical energy and vice versa [1]. According to the type of input, EMs can be classified into single-phase and three-phase machines [2], where the most common types of three-phase electric machines are synchronous (SM) and asynchronous machines (AM). Electrical machines (EM) are built from a stator, which is the static part of the EM consisting of phase-shifted coiled poles, and a rotor, which is the rotary part constructed depending on the tasks for which the machine is intended. AMs are most often used as motor-driven machines [3–5] in the automotive industry [6,7], construction [8], elevators [9], etc., while SMs are most often used to produce electricity in fossil fuel power plants [10–12] or in renewable energy plants, such as hydroelectric power [13,14] and wind power plants [15,16]. For energy to be transmitted from the producer to the final consumer, it must be of an alternating current (AC) [17]. The reasons for choosing AC over direct current (DC) are as follows [17,18]:

- Easy maintenance and change of AC voltage for transmission and distribution;
- AC transmission plant costs (switches, transformers, etc.) are much lower than equivalent DC transmission;
- The power plant produces AC power, so it is better to use AC than DC instead of converting;
- In the case of major faults in the network, it is easier to disconnect an AC system because the sinusoidal current tends to zero at a certain moment.

An SM has two characteristic parts, the armature on the stator and the excitation on the rotor where the armature winding (most often three-phase) is symmetrically distributed in slots around the circumference of the machine and indicates the part of the machine in which the changes of the magnetic flux induce a voltage [19]. DC flows through the exciting winding element, which is located on the rotor of the machine and creates the exciting flow, i.e., magnetic flux. Without the excitation current on the rotor, it would not be possible to produce the induced voltage on the armature windings of the synchronous generator, which means that there would be no electricity [19]. This type of excitation is used with synchronous generators that either have electromagnets on the rotor of the machine connected to a slip ring [20] or a rotor made with permanent magnets [21]. Estimation of the excitation current on the rotor is one of the key factors for the control of the SM. The excitation current maintains the efficient production of the induced voltage and is one of the most important factors while regulating the plant's power factor (maintaining the capacitive or inductive character of the power grid network) [22].

AC electrical machine loads draw reactive power from the electrical power grid. Reactive power is a disadvantage, i.e., a problem, because it causes overloading in the power grid, switches, transformers, and relays and, unfortunately, reactive power cannot be converted into useful, i.e., mechanical energy. To normalize the amount of reactive power in the power grid, SMs are used as reactive power compensators [23]. To achieve high-quality regulation and compensate for reactive power, it is necessary to regulate an SM's power factor parameter; this can be accomplished with proper regulation of the excitation current. The value of the excitation current dictates the operation mode of the synchronous motor (capacitive or inductive character) and affects the stability of the system [24].

The main problem is that most EM manufacturers do not provide enough information about their machines, which reduces the possibility of achieving highly efficient control. In general, manufacturers provide information on rated output power, rated maximum speed, rated input voltage, rated current, protection level, dimensions, and weight. However, a minority of manufacturers provide more valuable information, such as the speed-torque curve (most often at the request of the customer). Often, the nominal parameters of a synchronous motor are available and sufficient for designing the regulation, but the problem is the nonlinearity of the parameters, which is noticeable in an unadjusted operating environment (for example, high or low ambient temperature) or is not adapted for unique operating conditions (for example, speed or load torque) [25,26].

Apart from the nonlinearity, external and internal conditions affect an EM. The main problem is that there is no clear relationship between the parameters of the synchronous motor [27–30]. The parameters of an SM are mostly complex and nonlinear; thus, modeling SM parameters, such as excitation current, power factor, and load current, when the synchronous motor is running in a lagging, leading, or unity condition for reactive power compensation is a difficult task [24,27]. With the aim of improving the quality of SM modeling and a more precise estimation of parameters, many researchers use artificial intelligence (AI) estimation methods. The estimation of model parameters using techniques for linear systems has been perfected. However, an increasing number of systems require nonlinearities when increasing the dynamic range of high-performance equipment. Nonlinearities are usually ignored, but only under the assumption that linear system theories are sufficient for the retrieval of less accurate approximate solutions. For industrial applications, these solutions are most often within acceptable limits, but for applications in high-efficiency machines, linear systems are inadequate. There is therefore a need for further development of nonlinear systems, such as the synchronous motor. After defining the obstacles that are typical for highly coupled nonlinear system, such as synchronous motors, the authors of [27] used particle swarm optimization (PSO) to obtain a high-quality model with a low error rate that is robust and generally applicable to similar nonlinear systems. Various optimization algorithms, such as the evolutionary algorithm, ANN, artificial bee colony (ABC), immune method (IM), whale optimization method (WOM), particle swarm optimization (PSO) method, flower pollination (FP) method, cuttlefish optimization algorithm,

and genetic algorithm (GA), were used to optimize a permanent magnet SM (PMSM), and they are analyzed in detail in the state-of-the-art study in [25,31]. The parameters of the PMSM tend to change; in other words, there is an influence due to the nonlinearity of the parameters. This can occur due to the change in temperature or aging of SM, and for this reason, online techniques are used to identify more up-to-date and precise parameters [32].

In addition to potential optimization, AI is used to obtain improved waveforms at the output and reduce oscillations (such as output speed, torque, and current variations for three phases). For example, in [33], the authors used a fuzzy logic controller in combination with AI with certain conditions (in this case with 25 conditions and 49 conditions), which resulted in a much smoother rotation of the engine and thus less oscillation, which greatly contributed to the improvement of the system.

Using AI more precisely, fuzzy logic in combination with ANN represents an advanced method that is applied for AM control logic. AM is a nonlinear machine, where the influence of temperature, age, and additional vibration elements related to electromagnetism affect the operation of the machine. So, with this idea, the authors in [34], after mathematically modeling AM, defined a control strategy based on rotor flux, which ensures the robustness of the algorithm.

Regarding synchronous motors with electromagnets, the authors of [35] indicate the complexity and nonlinearity of the SM parameters. By applying the symbiotic organisms search (SOS) algorithm, gravitational search algorithm (GSA), ABC, and GA, the authors investigate the possibility of obtaining a high-quality algorithm with a small error of approximation, whereby the best results are achieved by SOS with a maximum error of 0.1703 A.

In the last decade, the implementation of AI algorithms has become an increasingly common strategy to solve parameter prediction, optimization, or control methods for electrical machines. Various studies are being carried out related to the optimization of excitation current losses. In the research of Kahraman et al. [24], the authors looked for an efficient solution to overcome the challenges in excitation current estimation. Implementing the k-nearest neighbor (kNN) algorithm led to optimal results of the excitation current, with an estimation error rate of 4.5% and standard deviation ($\sigma$) of 1.5%. Inter-turn faults in field winding can also affect the operation of the synchronous generator, which in this case is a turbogenerator. According to Guillen et al. [36], the authors used AI to obtain a model based on machine learning (ML) regression algorithms, where the excitation current was estimated using AI, and the algorithm estimated the field current, which was compared with the actual measured current in several lifetime fluctuations. From three different models (Potier, ASA, and SVR), SVR performed the best with mean average error (MAE) and root mean square error (RMSE) metrics of 0.448 and 0.009, respectively. With the use of artificial neural networks (ANN), it is possible to achieve a model for estimating the parameters of an SM; in [37], the authors use an adaptive artificial neural network (AANN). The purpose of this research was to estimate the excitation current and help designers with the modulation of the excitation current while developing sophisticated software with a low degree of programming and improving the efficiency of the classic ANN-based approach. The results were evaluated with an average error percentage, $\sigma$, and the arithmetic average of all error rates of 3.507958, 2.305857851, and 2.305858, respectively. Temperature monitoring of permanent magnets in SM, which is applied to the automotive industry, is a complex challenge. The heating of the machine can cause deterioration and impact the performance of SM itself. Monitoring and reacting to high-temperature variation values inside of an SM is a challenging task. Kirchgässner et al. [38] trained several ML models that are empirically evaluated based on their accuracy for the given task of predicting. The ML algorithms used for predicting high-dynamic latent magnet temperature profiles were as follows: linear ordinary least square regression, support vector regression, kNN, and neural networks. The best result on the test set for predicting the permanent magnet temperature was achieved with the convolutional neural network (CNN) with an $R^2$ of 0.99, MAE of 0.85 and MSE of 1.52; kNN and multi-layer perceptron

(MLP) achieved $R^2$ = 0.98, MAE = 1.32 and MSE = 3.20. Further proof that AI and ML can be applied in the modeling and simulation of complex phenomena is provided by surrogate models [39]. The authors presented a workflow for developing data-driven surrogate models including data generation with a physics-based simulation, designed experiments with the training data, and trained and tested the surrogates to compare ANN and the gradient boosting decision tree (GBDT) algorithm for estimation the torque behavior in SM with permanent magnets. The accuracy of the proposed model with the ANN algorithm was better, compared to the competitor GBDT, and was quite close to the finite element simulation in which the best result for negative root mean square (NMRSE) obtained with the non-hybrid model was 3.8%, while for the hybrid model it was 1.76%. Mukherjee et al. [40] estimated the speed and torque of a PMSM using tree-based algorithms. Several tree-based algorithms were used where the fine tree algorithm achieved a much lower error rate compared to the other tested algorithms, namely medium tree and coarse tree. The RMSE for the fine tree algorithm was 0.029224 and 0.052538 for the prediction of speed and torque, respectively. Traue et al. [41] developed a reinforcement learning environment toolbox for intelligent electric motor control using the open-source Python package. The obtained model showed better results in control tasks compared to conventional methods. Li et al. [42] conducted a detailed analysis for the optimization of electromagnetic devices. By using various algorithms such as support vector machines, MLP, k-NN, and CNNs, the authors proved that ML algorithms have greater robustness, high speed, and accuracy by applying a fitting algorithm in different scenarios. Bayindir et al. [43] showed the importance of the excitation current in synchronous motors in the case of reactive power compensation. By using the k-NN classifier and load current, power factor, power factor error, and change of excitation current as input values, the following results are obtained for MAE, mean absolute percentage error (MAPE), and NRMSE: 0.059, 13.146, and 8.167, respectively. The given metrics are referred to as three tuple input values, which were as follows: load current, power factor, and change of excitation current, which showed the importance of certain parameters when modeling the AI algorithm and the viability of using AI for optimizing and predicting the parameters in EM.

Considering the high accuracy rate of AI algorithms in prediction and regression, the following hypothesis questions are raised regarding SM:

- Is it possible to estimate the excitation current of SM using AI algorithms with a high precision rate and a small evaluation error?
- Is it possible to optimize the model and confirm the obtained results with 5 k-fold cross-validation using the randomized hyperparameter search?
- Which algorithm provides the best results with the possibility of implementation in a real-life situation?

This research was conducted to obtain an optimal AI algorithm that, with its properties and characteristics, can estimate the excitation current of the SM (in this case, synchronous motor drive), which contains nonlinear characteristics and phenomena. The best AI algorithm is determined to predict the future values of the excitation current with a high accuracy, small standard deviation error ($\sigma$), and almost non-existent regression metrics error. By comparing the best algorithm with the related work, i.e., comparing the extreme gradient boosting regressor with small hyperparameter variations and robustness, a solution for the given problem can be obtained. In this way, precautions can be taken for plants with these machines and additional challenges can be prevented in time. The structure of this paper, along with the introduction, is divided into the following four sections: Materials and Methods, Results and Discussion, and Conclusions. The Materials and Methods section describes the entire dataset, including the input values along with targeted value, an analysis of the dataset distribution using histogram bar plots is provided, and the SM used in the creation of the dataset is described. The analysis of the dataset used for training ML algorithms is shown, and the algorithms used for the prediction of the excitation current are stated alongside the hyperparameters used in randomized hyperparameter search. All models were validated with 5 k-fold cross-validation. The Results and Discussion section

shows the results obtained from the selected/used AI algorithm, the parameters used for research, and the training of the best obtained AI model. The achieved results were described and we elaborate on which algorithm provided the best results, and which one was the worst according to the obtained metrics. In the Conclusion, the final results of this research are outlined, including which algorithm had the best regression performance for the given challenge, and the initial hypotheses questions are answered.

## 2. Materials and Methods

In the Materials and Methods section, the SM used for data collection is presented. The dataset is described as well as the input and output variables. The input data and statistical analysis before the actual use of ML algorithms are presented. Additionally, the potential challenges are indicated, and the importance of AI implementation is highlighted.

### 2.1. Potential Challenges When Modeling a Synchronous Motor

In modern EM modeling, the contribution of Štumberger et al. [44], Kron [45] is important, in which the general theoretical background for EM modeling is outlined. The generality of Kron's tensor-based approach diminishes when using matrices. By using matrices, EMs are mostly treated as magnetic (deterministic) linear systems, while nonlinear properties are ignored. However, to create positive ratios between measured and calculated results, nonlinear theory must be included in the model of the SM.

In addition to neglected factors when modeling a synchronous machine, there are several losses in SM that make the machine complex and nondeterministic and require additional power (iron losses, winding losses, and ventilation losses) [46]. Of course, these losses depend on the situation in which the machines are used; this makes it difficult to estimate parameters with regular measuring methods. Certain calculations can be made to gather information; however, due to the complexity of thermal effects in EM, numerical calculation methods such as electromagnetic and thermal finite element modeling (FEM) or computational fluid dynamics (CFD) are only used for the specific analysis of EM (part of the stator core, for example). A complete thermal analysis would require full CFD/FEM coupling. As for the modeling of the SM, the turn insulation in the field winding in the case of the silent pole synchronous machine (generator) is thin, so it is often neglected in calculations. However, surface insulation is of similar dimensions, almost equal to the surface of the conductor, so it would affect the result of the heat calculation. On the other hand, the calculation of the temperature field of a synchronous machine with silent pole rotor insulation in terms of the insulating layer is often neglected; even with little insulation there is a temperature drop, which can affect the modeling of the EM itself [47].

This subsection highlights the limitations of calculations when modeling or estimating EM. This highlights the importance of using AI algorithms for the optimization or estimation of synchronous machine parameters. AI algorithms are based on the data contained in the dataset, and the more data the dataset (algorithm) has, the greater the robustness and readiness, as well as the accuracy of the obtained model. In this work, the the parameters of the used synchronous motor and the statistical analysis and distribution of data for input into the AI model are described.

### 2.2. General Information about SM

As presented in the introduction of this paper, all AC machines consist of a stator and a rotor. The main difference between AM and SM is the speed of rotation of the rotor, which in the case of SM, as the name indicates, rotates synchronously. This means that there is no delay in the rotating magnetic field of the rotor relative to the stator. In terms of the design of the rotor, SM has two versions. The first one is the salient pole [48] shown in Figure 1, and the other one is the non-salient pole (round/cylindrical) rotor [49] shown in Figure 2.

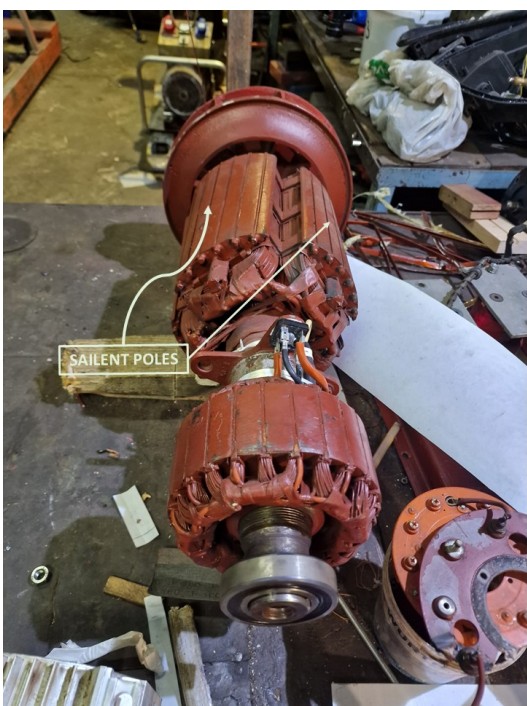

**Figure 1.** Salient-pole type rotor SM.

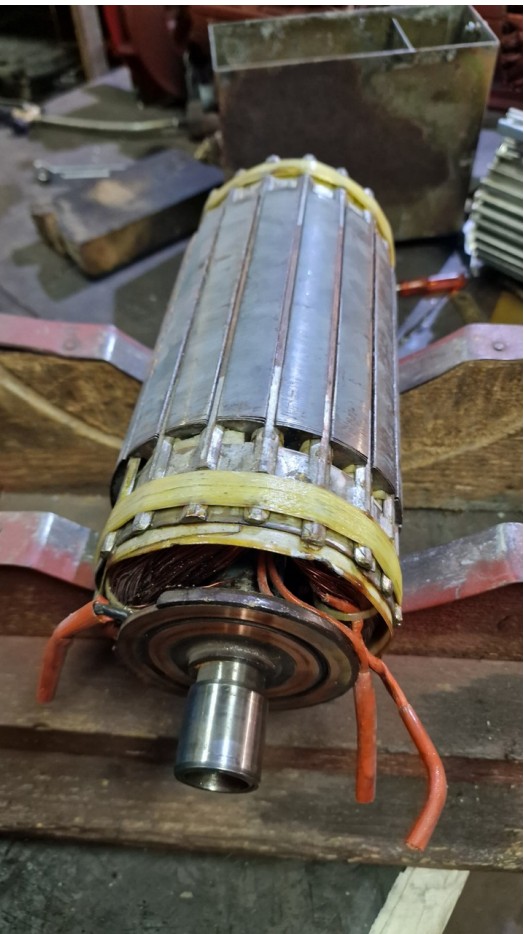

**Figure 2.** Round rotor type SM.

Figure 3 shows the section of the rotor and stator (armature) inside the SM. The figure shows the position of the rotor field winding for both configurations and the position of the polarization for both designs.

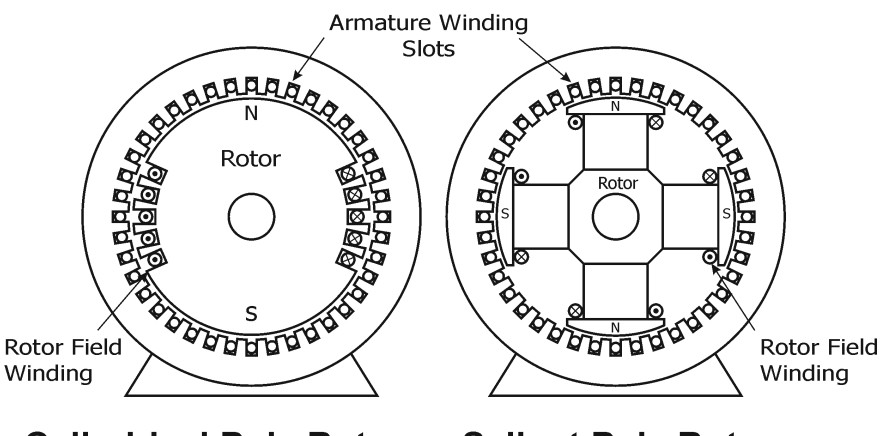

**Cylindrical Pole Rotor  Salient Pole Rotor**

**Figure 3.** Representation of the salient pole and cylindrical pole rotor type SM.

The overall configuration is shown in Figure 4, which shows each component of the SM operation. In Figure 4, every part of SM is visible. To create a rotating magnetic field with a phase shift of 120 degrees, a three-phase voltage is required, with each phase marked as a, b, c and a', b', c'. The dotted dashed lines indicate the position of the phased coils, while the hatched part indicates the location of the conductors located in the armature SM.

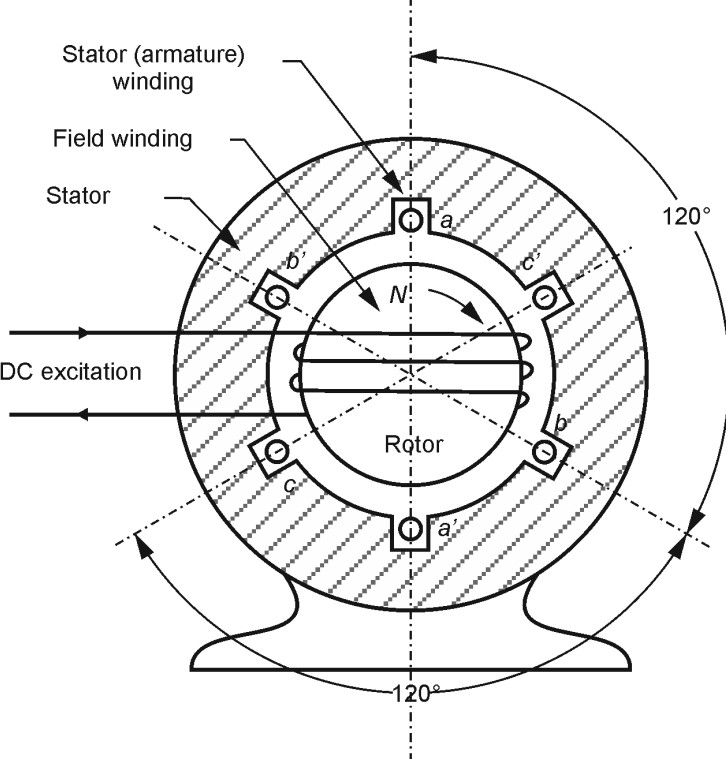

**Figure 4.** The basic operation of SM.

The working principle of SM is based on the establishment of a magnetic field that rotates at a synchronous speed, as follows:

$$n = \frac{120 \cdot f}{p}. \tag{1}$$

As shown in Equation (1), the speed of the rotating magnetic field of the stator depends on two factors f which are the frequency of the rotating magnetic field and p the number of poles. If the speed of the rotor (magnetic field caused by the DC component) [50], and the speed of the rotating magnetic field of the rotor are equal, under the condition that there is no load torque, both magnetic fields will tend to align against each other. When the load is applied, the rotor lags a little, i.e., "slips" by a certain degree, but at the same time adheres to the revolutions of the rotating magnetic field [49]. The achievement of the highest torque is due to the relationship between the magnetic field of the rotor and the stator, which attempt to maintain a value of 90 degrees, as can be seen from Figure 5.

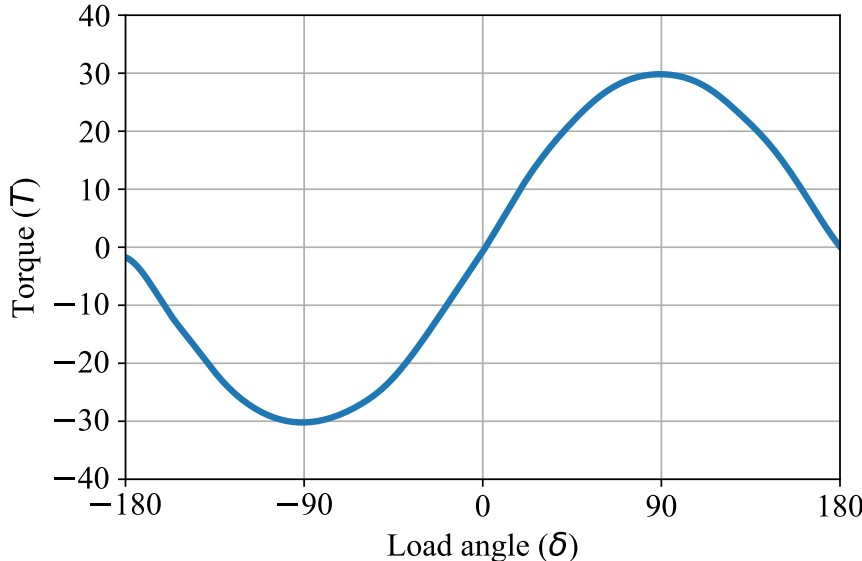

**Figure 5.** The effect of load angle on produced torque.

Figure 5 shows the production of torque (T) under the effect of load angle ($\delta$); it is visible that when SM has an angle of 90 degrees, the maximum T is produced [49].

$$T = T_{max} \cdot \sin(\delta). \tag{2}$$

In addition to displaying the maximum torque at a rotor angle of 90 degrees [51], it is possible to calculate the current torque using Equation (2), where:

- T is the calculated torque;
- $T_{max}$ is the maximum torque for SM;
- $\sin(\delta)$ is the sinus function of load angle.

### 2.3. Operation Conditions and Dataset Collection

So far, the general view of SM has been described, but it is also important to describe the working conditions of the synchronous motor in which the measuring of the parameters is performed. The working conditions of the SM (synchronous motor regime of work) that were measured are shown and described in Table 1 [52].

**Table 1.** Operating properties of experimental synchronous motor used in this research.

| Condition | | | |
|---|---|---|---|
| Star connected motor (Y) voltage [V] | Star connected motor (Y) current [A] | Triangle connected motor (Δ) | Star connected motor (Y) current [A] |
| 400 | 5.8 | 231 | 10 |
| Power factor (cos ∅) | | | |
| 0.8 | | | |
| Apparent power (kVA) | | | |
| 4 | | | |
| Revolutions per minute (rpm) | | | |
| 1000 | | | |

An auxiliary motor is used to drive the synchronous motor in the test rig. In the star connection (Y), the motor has a voltage of 400 V with a 5.8 A current. In the delta connection (Δ), the values are different, and the voltage amount is 231 V with a current of 10 A. The values for the power factor (cos ∅), apparent power, and revolutions per minute (rpm) are the same, at 0.8, 4 kVA, and 1000, respectively [52].

The data collection procedure for the synchronous motor is shown in Figure 6 of this investigation, and it was conducted in the following four steps: firstly, the motor was used as an auxiliary motor in a laboratory environment, while a series rheostat was used to obtain information about the DC supply variable in the field circuit ($I_f$). Secondly, using the AC voltage, the stator windings of the synchronous motor are powered to achieve a synchronous speed. In the third step, DC voltage is applied to the field winding of the synchronous motor, and the motor enters synchronism. In the last step, synchronous speed is realized, and the field current is modified to the minimum value (using a serial rheostat connected in series to the field circuit.) After performing all the steps, the motor takes the minimum current value from the power supply and obtains the maximum efficiency with a stable power factor [52].

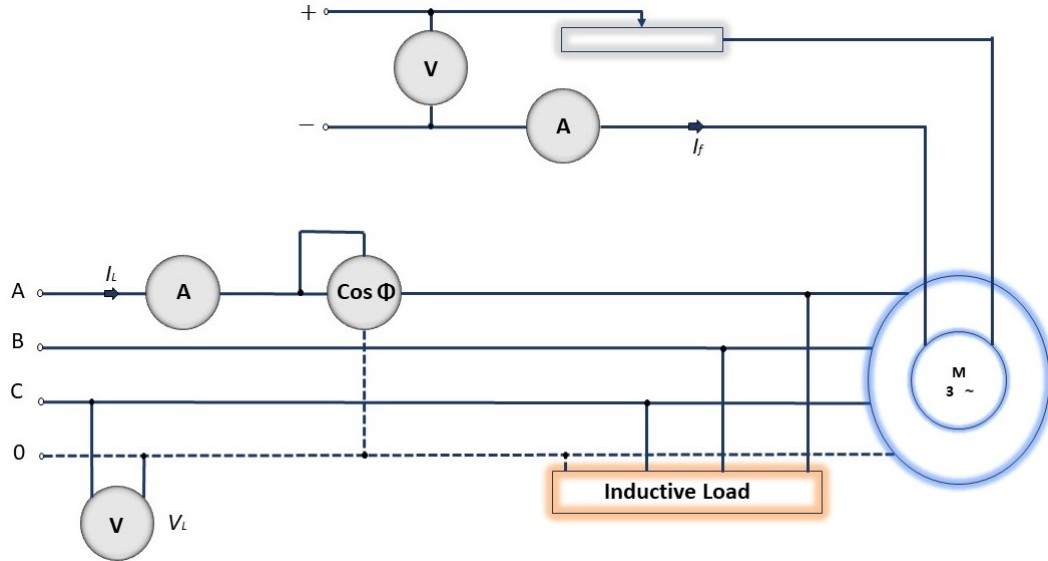

**Figure 6.** The scheme of work for the experiment with an SM [52].

*2.4. Dataset Statistical Analysis*

The previously described data are sufficient for the description of the dataset collection process and the dataset creation; however, it is also recommended to additionally analyze

the dataset with the aim of obtaining better insight into the state of the variables and their interrelationships. In Table 2, there are four specific values that were analyzed for each individual variable, namely the mean value, minimum value, standard deviation, and maximum value.

**Table 2.** Dataset analysis used in research.

| | $I_y$ | PF | e | $d_f$ | $I_f$ |
|---|---|---|---|---|---|
| Mean value | 4.499 | 0.825 | 0.174 | 0.350 | 1.530 |
| Minimum value | 3.0 | 0.650 | 0.0 | 0.037 | 1.217 |
| Standard deviation | 0.896 | 0.103 | 0.103 | 0.180 | 0.180 |
| Maximum value | 6.0 | 1.0 | 0.350 | 0.769 | 1.949 |

The mean value represents the mean value in each variable, the minimum represents the lowest measured value, while the maximum represents the highest measured value. It is especially important to analyze the standard deviation of the data. This is performed to consider the possibility of implementing the pre-processing methods for the input variables of the dataset, which were $I_y$, PF, e, and $d_f$ for the estimation of the target variable $I_f$. Table 2 shows that the deviation of the data is not too high, while the mean value of the data is mostly above the ideal mean value (concerning the minimum and maximum value of the individual variable). This leads to the conclusion that it is not necessary to perform data pre-processing. Further on, the training of the AI model is performed using the original dataset values for each variable.

Furthermore, in addition to defining operating conditions and statistical analysis of data, it is necessary to consider the distribution of data in each variable, so starting from $I_y$ shown in Figure 7, the distribution of data for each variable was analyzed.

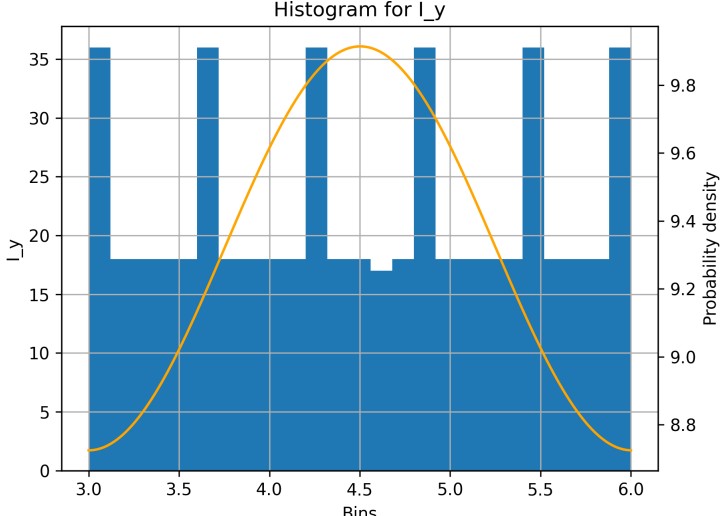

**Figure 7.** Data distribution/histogram for Load current in the dataset, the histogram consists of an analyzed parameter as the number of inputs with the given value.

Figure 7 shows a histogram plot of the data distribution for the $I_y$ parameter. The blue bars represent input bins while the orange curve represents the best-fitted distribution for the given data. In statistics and probability theory, this form of data distribution belongs to Von Mises lines, also known as a circular normal distribution, and is a continuous probability distribution of the circle. It is a close approximation of the wrapped normal distribution, which is the circular analogy of the normal distribution. The circular analogy can be seen

by the repeating peaks of individual values (interval maximum values of individual bar plots). The Von Mises probability density function can be presented as follows:

$$f(x \mid \mu, \kappa) = \frac{e^{\kappa \cos(x-\mu)}}{2\pi I_0(\kappa)}, \tag{3}$$

where is:

- $x$, angle of the density function;
- $\mu$ is a representation of measure location (the given cluster distribution around $\mu$);
- $\kappa$ is a representation of the measure concentration;
- $I_0(\kappa)$ is the modified Bessel function with order zero.

Figure 8 shows the data distribution related to PF and e. Based on the entered data, the best possible distribution function corresponds to the Three-Parameter Kappa distribution. In statistics, K-distribution or Kappa 3 distribution is a family of continuous probability distributions that consists of three parameters and is constructed by combining two gamma distributions. The probability density function is presented as follows:

$$f(x, a) = a(a + x^a)^{-(a+1)/a}, \tag{4}$$

for $x > 0$ and $a > 0$ and $a$ and $x$ are the shape parameters of the function.

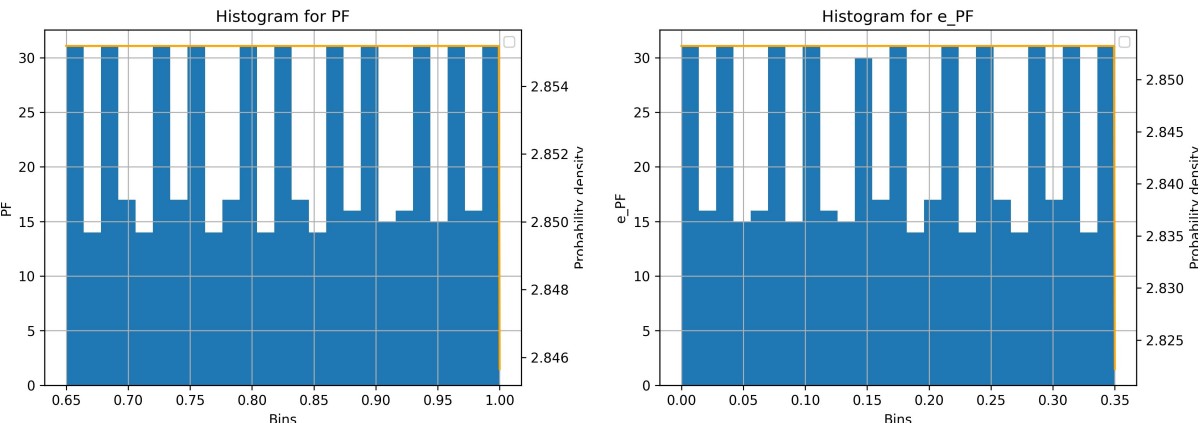

**Figure 8.** Data distribution/histogram for power factor PF and power factor error e in the dataset, the histogram consists of an analyzed parameter as the number of inputs with the given value.

In the case of an $I_f$ and $d_f$ of the SM, the distribution of data is slightly different than in the previous cases. By observing Figure 9, it is evident that the distribution of the data resembles a standard Gaussian distribution; however, due to the shift of the data for a certain parameter, the best-fitted distribution is Johnson's SB distribution. From Figure 9, it is evident that the highest ratio of data is at smaller values. The probability density function can be represented mathematically as follows:

$$f(x, a, b) = \frac{b}{x(1-x)} \phi \left( a + b \log \frac{x}{1-x} \right), \tag{5}$$

where:

- $x$, $a$, and $b$ are real scalars;
- $b > 0$ and $x \in [0, 1]$ is the probability density function of the normal distribution;
- $\phi$ is the cumulative distribution function of the normal distribution.

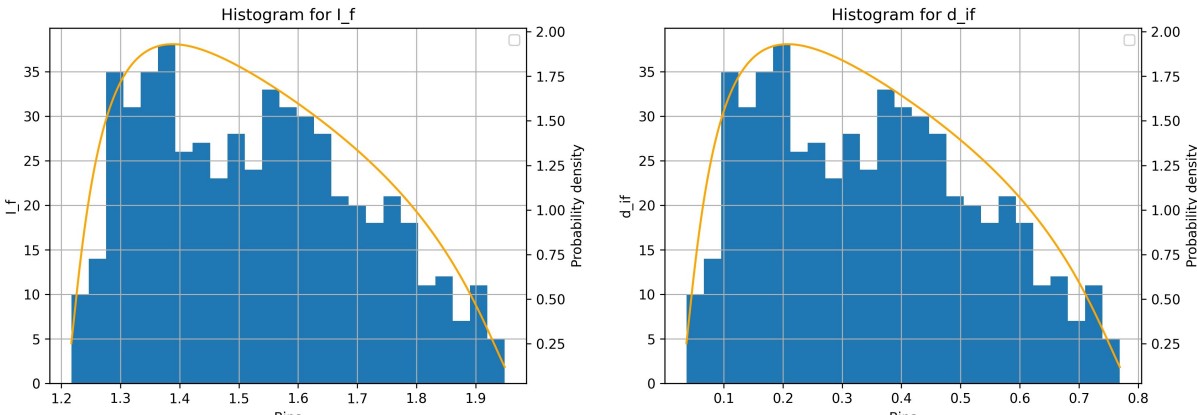

**Figure 9.** Data distribution/histogram for excitation current $I_f$ and changing of excitation current $d_f$ of synchronous machine; the histogram consists of an analyzed parameter as the number of inputs with the given value.

At the given distribution of data, shown in the histograms in Figures 7–9, the first logic attempt is to train the ML model without data pre-processing. The given dataset has favorable conditions for training ML algorithms, and in this paper, the prediction of $I_f$ is performed with the non-preprocessed dataset.

### 2.5. Research Methodology

This section describes the research methodology of the various AI algorithms that were used in this investigation and their hyperparameters with the given table values of the selected hyperparameter. Linear models, ensembles, and even more complex ones such as artificial neural networks (ANN) and tree-based regression methods of value prediction were considered. Hyperparameter optimization was performed using randomly selected hyperparameters and by training the model using the grid-search (GS), and was validated using 5 k-fold cross-validations. During this research, the Python programming language and scikit learn library (known as sklearn) version 1.0.2 were used. The aforementioned Python library holds all of the used machine learning algorithms (the exception was the extreme gradient boosting regressor), whereby an initial investigation was conducted using default hyperparameters for individual algorithms. After the initial investigation, the hyperparameter optimization was conducted using random values that are presented in each table for every algorithm used, where the used algorithms are as follows:

- Extra trees regressor (ETR);
- Elasticnet regressor (EN);
- K-nearest neighbor regressor (k-NN);
- Linear regressor (LR);
- Random forest regressor (RFR);
- Ridge regressor (RR);
- Stochastic gradient descent regressor (SGD);
- Support vector regressor (SVR);
- MLP regressor;
- Extreme gradient boosting regressor (XGBoost).

Each algorithm will be described in a separate subsection where the hyperparameter values used for training the model will be shown. During the investigation, the following two search methods were applied: the initial search with default hyperparameters, and then the randomized hyperparameter search. After the initial search, all the algorithms were optimized using GS and cross-validation.

### 2.5.1. Extra Trees Regressor

ETR belongs to the group of ML ensemble methods that combines predictions from multiple decision trees. It belongs to a random forest algorithm although it is based on a simpler approach where each member is a decision tree. In this case, the mentioned algorithm often results in a similar or better result than the base algorithm, which is the random forest algorithm [53]. ETR trees differ from the usual decision trees in terms of the principle of building the tree itself. The search for the best division for separating the hub samples into two rough ones is drawn for a random division of the selected features, i.e., max_features, and the best division among them is selected [54]. The varied hyperparameters of the algorithm are visible in Table 3 and are varied based on previous research according to [55–57].

**Table 3.** An overview of the varied hyperparameters used for the ETR.

| Parameter Name | Minimum Value | Maximum Value |
|---|---|---|
| n_estimators | 1000 | 10,000 |
| criterion | squared error, friedman_mse | |
| max_depth | None | |
| min_samples_split | 2 | 10 |
| max_features | auto, sqrt, log2 | |
| random state | 0 | 63 |

The criterion is a quality measurement of split data. For this kind of decision tree algorithm, supported criterion inputs are squared_error for the mean squared error (MSE) and friedman_mse for the improved Friedman mean squared error. The main difference between these two criteria are that the Friedman MSE calculates the impurity of the current node which leads to a reduction in the leaf impurity. The number of trees that ETR produces is regulated with n_estimators. Max_depth indicates the maximum depth of the tree. It was set to none where the main reason was to remove all of the leaf impurities from the tree. max_features is the number of considered features when looking for the optimal split. Input values can be sqrt, log2, integer, or float number. Random_state controls the randomness of the instance of bootstrapping samples while building trees and the feature sampling for consideration when looking for optimal node split, and draws the splits for max_features. The lowest value of the sample number for splitting the internal node is defined with min_samples_split. If the given value is an integer, then min_samples_leaf is considered as the minimum number; in the case of the float value, the min_samples_split is considered as a fraction.

### 2.5.2. Elasticnet Regressor

EN belongs to the group of linear models of the scikit library where the regression process uses the penalties from both lasso and ridge techniques. This technique provides a learning spares model, where several weights are not 0 as is the case in lasso, while still maintaining the regularization characteristic from the ridge. The combined technique from lasso and ridge regression methods helps to overcome the deficiency and improve the statistical regulation model [54,58]. During the investigation of this AI algorithm, the parameters shown in Table 4 were varied according to [59,60].

**Table 4.** An overview of the varied hyperparameters used for the EN.

| Parameter Name | Minimum Value | Maximum Value |
|:---:|:---:|:---:|
| alpha | 0.1 | 10 |
| l1_ratio | 0.1 | 10 |
| max_iter | 1000 | 10,000 |
| selection | random, cyclic | |

Alpha constantly multiplies the penalty terms if its value is set to 0.0; then, the mathematical meaning of this parameter is equivalent to a linear ordinary least square regression. The elasticnet l_1_ratio is the mixing parameter between 0.0 and 1.0. The maximum number of iterations is max_iter set to 10,000. The selection parameter is the coefficient updated for every iteration by choosing the right coefficient of selection, and the results often converge faster.

### 2.5.3. K-nearest Neighbour Regressor

The k-NN algorithm is one of the decision tree supervised ML algorithms based on determining the local minimum of the target function that is used to learn the unknown function with precision and accuracy. The decision system essentially calculates the distance between data points in space. It relies on observable data similarity and near-distance metrics for accurate prediction generation [38,54]. When modelling k-NN based on research in [35,54,61], the parameters shown in Table 5 were varied.

N_neighbours is the number of neighbor data in the decision tree. There are three ways in which the weights of the prediction function are distributed, which are uniform, inverse, and callable. The computation of the closest neighbor is achieved with an algorithm. The eaf_size can be used in BallTree or KDTree. It can be used to determine the memory of a given leaf of the tree as well as speed up the construction of the building process.

**Table 5.** An overview of the varied hyperparameters used for the k-NN.

| Parameter Name | Minimum Value | Maximum Value |
|:---:|:---:|:---:|
| n_neighbours | 1 | 1000 |
| weights | uniform, distance | |
| leaf_size | 1 | 1000 |
| algorithm | auto, ball_tree, kd_tree, brute | |
| p | 2 | 50 |

### 2.5.4. Linear Regressor

LR is better known as the ordinary least squares method. The method consists of establishing a correlation between the dependent variable and the independent variable, where the best approximation fits a straight linear line [62]. The variation of parameters is presented in Table 6.

**Table 6.** An overview of the varied hyperparameters used for the LR.

| Parameter Name | Minimum Value | Maximum Value |
|:---:|:---:|:---:|
| fit_intercept | True, False | |
| normalize | True, False | |
| positive | True, False | |

Fit_intercept calculates the intercept for the given model. Normalize is used for normalization of the input parameter, and positive is used if the True forces modulation coefficients are positive.

### 2.5.5. Random Forest Regressor

The RFR algorithm belongs to the class of machine learning algorithms that, in combination with random decision trees, trains a model on sub-datasets [63]. The use of multiple trees contributes to the stability of the algorithm and reduces the variance of the result itself. Each tree created by this algorithm uses a different sample of input data; furthermore, different features are taken into account at each node where, after training each tree, the mean value of the unique result is calculated [63,64]. The varied hyperparameters are shown in Table 7 and are based on [55,65,66].

**Table 7.** An overview of the varied hyperparameters used for the RFR.

| Parameter Name | Minimum Value | Maximum Value |
| --- | --- | --- |
| n_estimators | 100 | 5000 |
| criterion | squared_error, absolute_error, Poisson | |
| max_features | sqrt, log2 | |

Given that the ETR in section 2.5.1 originated from the basic RFR, the parameters that were varied in this investigation will not be described in detail. When investigating this algorithm, the guiding principle was that differences between the two algorithms exist in terms of ETR [67]. In this case, RFR uses bootstrap replicas, i.e., it uses subsamples of the input data and replaces it, whereas ETR uses the original complete samples. In addition, RFR uses an optimal node split point, while ETR takes a random one.

### 2.5.6. Ridge Regressor

RR belongs to the linear model method of the scikit-learn python library. The specialization of RR lies in the fact that it analyzes multiple regressions that are multicollinear. Because of the complex science behind this ML algorithm, it is not the most used algorithm; however, because of its regularization methods, it is sometimes the best fit. RR is a linear least squares method with an additional regularization parameter. The selection of the varied parameters presented in Table 8 was carried out according to previous [68–70] articles on related issues.

**Table 8.** An overview of the varied hyperparameters used for the RR.

| Parameter Name | Minimum Value | Maximum Value |
| --- | --- | --- |
| alpha | 1.0 | 100.0 |
| max_iter | 1000 | 50,000 |
| tol | $1 \times 10^{-5}$ | $1 \times 10^{-1}$ |
| fit_intercept | True, False | |
| solver | svd, cholesky, lsqr, sparse_cg, sag, saga | |

The parametersalpha and max_iter are explained in the previous subsections, tol indicates the precision of the solution, while the novelty is the solver where the solver is the principle of model optimization.

### 2.5.7. Stochastic Gradient Descent Regressor

The SGD regressor belongs to the group of linear models of the scikit-learn python library. It is a generic optimized algorithm that can find a high-quality and optimal solution for a wide range of situations. The idea for altering the parameters is an iterative process

of selecting hyperparameters to reduce the cost function. The proportions of the step are one of the most crucial parameters determined by varying the learning rate, which in the case of setting too small a value can lead to slow convergence and the opposite of that can exceed the optimal value [71,72]. In line with to [73,74], the hyperparameters in Table 9 are as follows.

**Table 9.** An overview of the varied hyperparameters used for the SGD.

| Parameter Name | Minimum Value | Maximum Value |
| --- | --- | --- |
| alpha | 0.0001 | 10.0 |
| max_iter | 1000 | 10,000 |
| validation_fraction | 0.15 | |
| power_t | 0.1 | 0.5 |
| learning_rate | invscaling, optimal, constant | |
| shuffle | True, False | |
| l1_ratio | 0.0001 | 0.5 |

The validation set is a proportion of training data used for early stopping defined with validation_fraction. The power_t parameter is an exponent for the invscaling learning_rate module which defines the result of the value of the learning rate parameter. The role of the shuffle parameter is to shuffle data points for each training epoch. The elasticnet mixing parameter is varied with the l1_ratio, and its value is in the range from 0 to 1.

### 2.5.8. Support Vector Regressor

SVR is an ML regression algorithm that is compatible with both linear and polynomial regression. This algorithm works on the principle support vector machine (SVM), but in the case of SVR, it differs in the way that SVM is a classifier for the prediction of discrete categorical values, while SVR is a regressor used for continuous order predicting variables [75,76]. In this investigation, several hyperparameters are varied, as shown in Table 10.

**Table 10.** An overview of the varied hyperparameters used for the SVR.

| Parameter Name | Minimum Value | Maximum Value |
| --- | --- | --- |
| kernel | linear, poly, rbf, sigmoid | |
| degree | 1 | 10,000 |
| gamma | scale, auto | |
| coef0 | 0.0 | 10.0 |
| tol | $1 \times 10^{-5}$ | $1 \times 10^{-10}$ |
| C | 0.5 | 20.0 |
| epsilon | 0.05 | 20.0 |
| max_iter | 100 | 10,000 |

According to previous research, [77–79], the varied hyperparameters are as follows. The kernel hyperparameter specifies a data-mapping function. The degree is the degree of the polynomial kernel function, and gamma is the kernel coefficient but only in the case of rbf, poly, and sigmoid functions. Coef0 is an independent term in kernel function and can only be significant in the poly or sigmoid kernel function, C is a regularization parameter where the amplitude of regularization is inversely proportional and must be strictly positive. The last hyperparameter is epsilon, which defines the margin of tolerance.

### 2.5.9. Multi-Layer Perceptron Regressor

An MLP is a feed-forward ANN that generates a set of defined outputs from an input value. Specifically, it is characterized by a node that is connected to several layers as a direct link between input, hidden, and output layers where a backpropagation logic is used for training the neural network itself. Its design allows it to predict any continuous function and it can solve nonlinearly separable problems. In this investigation, several hyperparameters are modified according to research shown in [54,80–82].

Table 11 shows the varied hyperparameters used in this investigation. In addition to the previously described hyperparameters, such as olver, alpha, power_t, max_iter, and tol, new parameters unique to the MLP ML algorithm have been changed. Activation stands for the activation function which decides whether a neuron contained in MLP should be activated or left in stasis. Using simple mathematical operations, it determines the importance of the selected neuron in the ANN network for the prediction process. Hidden_layer is a number of hidden layers inside the MLP ANN, and the solver is the algorithm specified for weight optimization along all of the nodes contained in MLP.

**Table 11.** An overview of the varied hyperparameters used for the MLP.

| Parameter Name | Minimum Value | Maximum Value |
| --- | --- | --- |
| hidden_layer_sizes (2,3 and 4 hidden layers) | 5 | 600 |
| activation | tahn, relu, identity, logistic | |
| solver | adam, lbfgs | |
| alpha | 0.02 | 0.5 |
| power_t | 0.5 | 2.0 |
| max_iter | 1000 | 10,000 |
| tol | $1 \times 10^{-5}$ | $1 \times 10^{-1}$ |
| max_iter | 1000 | 10,000 |

### 2.5.10. Extreme Gradient Boosting Regressor

XGBoost is an AI algorithm that is efficient and scalable for implementation based on the gradient boosting framework. The algorithm includes a linear model solver and a tree-based algorithm that supports several objective functions, including regression, classification, and ranking. Its special features are the possibility to regularize learning, the gradient tree boosting functionality, and the ability to shrink subsample columns [83,84]. Due to its performance and previous research, it was taken into account during this investigation. So, the varied hyperparameters used in this research are shown in Table 12.

**Table 12.** An overview of the varied hyperparameters used for the XGBoost.

| Parameter Name | Minimum Value | Maximum Value |
| --- | --- | --- |
| learning_rate | 0.2 | 0.02 |
| max_depth | 6 | 64 |
| min_child_weight | 1 | 10 |
| gamma | 0.0 | 1.5 |
| colsample_bytree | 0.1 | 1.0 |
| max_delta_step | 0.0 | 1.5 |

Learning_rate in XGBoost is slightly different than in the previously mentioned algorithms. In this case, the mentioned hyperparameter represents a reduction for each step that the algorithm performs. Min_child_weight is a parameter that defines the minimum

sum of instance weights needed for generating a child. The pseudo-regularization hyper-parameter specific to gradient boosting and XGBoost is gamma. It is vital because the higher the gamma is, the better the regularization process. Subsampling is performed using colsample_bytree, which takes a portion of the defined dimensions. Max_delta_step is the maximum step allowed in the algorithm for each tree's weight estimation.

## 3. Results and Discussion

In the results section, the results of this research are presented. The following evaluation metrics are presented for each algorithm: The coefficient of determination ($R^2$), MSE, and MAPE. When defining the results, the best-performing algorithms were selected, which were additionally evaluated using the $\sigma$ for all three individual metrics.

The first part of the research refers to the defined algorithms in Section 2.5 from Sections 2.5.1–2.5.10. Algorithms were trained on default parameters from the scikit-learn Python library. Each algorithm model was evaluated with three previously defined metrics, and the results can be found in Table 13.

**Table 13.** Table of the results and evaluation metrics of all algorithms used in this research for default hyperparameters.

| Regressor Name | $R^2$ | MSE | MAPE |
|:---:|:---:|:---:|:---:|
| Extra Trees | 0.9795 | 0.0006 | 0.0166 |
| ElasticNet | 0 | 0.0297 | 0.1442 |
| k-Nearest Neighbour | 0.9517 | 0.0015 | 0.0294 |
| Linear | 0.8875 | 0.0035 | 0.0465 |
| Random Forest | 0.9909 | 0.0003 | 0.0121 |
| Ridge | 0.9012 | 0.0028 | 0.0423 |
| Stochastic Gradient Descent | 0 | 0.0402 | 0.1611 |
| Support Vector Machines | 0.8662 | 0.0041 | 0.0506 |
| Multi-layer Perceptron | 0.8534 | 0.0052 | 0.0585 |
| Extreme Gradient Boosting | 0.99433 | 0.0001 | 0.0074 |

Table 13 shows the results of the AI algorithm used in this research with default hyperparameters for each algorithm. The goal of the initial research was to determine algorithms suitable for estimating the excitation current of SM. This part of the research indicates the oscillations of the results for each algorithm, which can be seen from Table 13. Several acceptable algorithms satisfy the given tasks including the tree-based algorithms, which had the best results in the initial research. Regarding $R^2$, the best result is achieved with XGBoost, of 0.9943, followed by 0.99 with RFR. Furthermore, results above 0.9 are given by RFR, ETR, and RR, as shown in Figure 10.

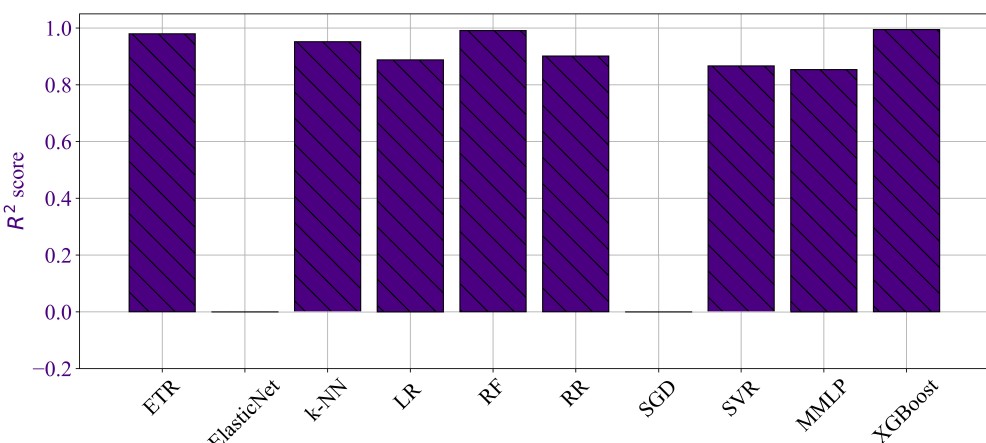

**Figure 10.** The $R^2$ results from the initial investigation.

Regarding other evaluation metrics, XGBoost again has the best result; in this case, the smallest MSE and MAPE are 0.0001 and 0.0074, respectively. The second-best algorithm with the best performance is RF with 0.003 and 0.0121 MSE and MAPE. The third best algorithm is ETR with 0.006 and 0.016. Other algorithms also have a low error rate in both metrics, although with at least one potential higher than the mentioned algorithms, as shown in Figure 11.

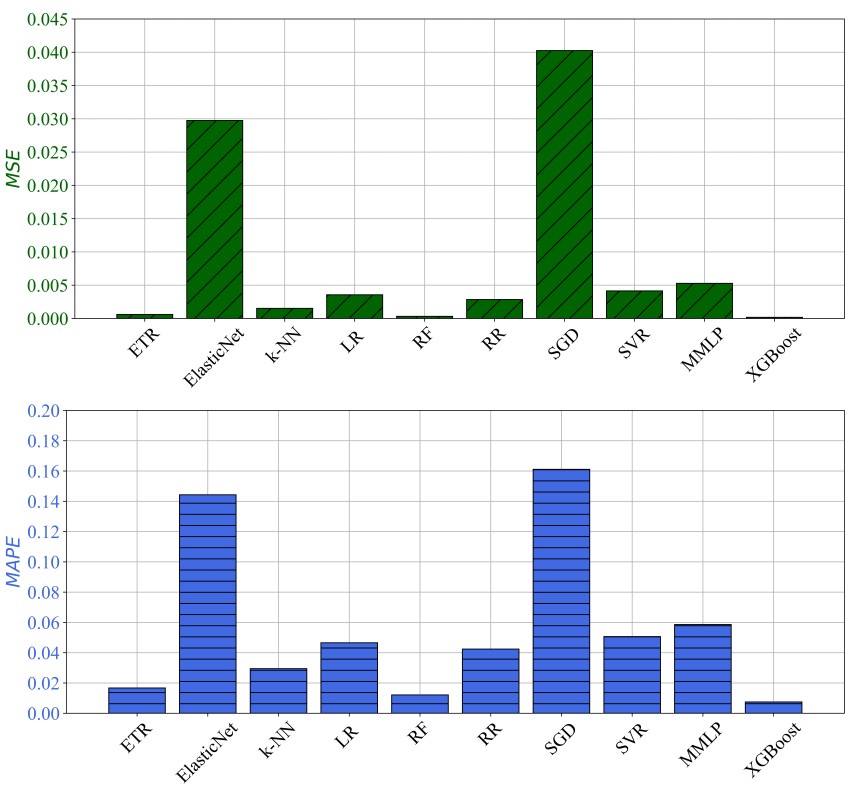

**Figure 11.** The MSE and MAE results from the initial investigation.

The most frequent problem faced while training ML algorithms is overfitting of the training data. Overfitting can be avoided by using the cross-validation method, as is shown in the second part of this investigation. Parameters presented in individual tables in the subsections from Sections 2.5.1–2.5.10 were used, and the optimization was achieved using a randomized hyperparameter search method with 5 k-fold cross-validation for each algorithm. The results of optimized and cross-validated algorithms can be seen in Table 14.

**Table 14.** Table of results and evaluation metrics of all algorithms used in this research for varied hyperparameters and cross-validated models.

| Regressor Name | $\overline{R^2}$ | $\overline{MSE}$ | $\overline{MAPE}$ |
|---|---|---|---|
| Extra Trees | 0.9784 | 0.0006 | 0.0164 |
| ElasticNet | 0 | 0.0297 | 0.1442 |
| k-Nearest Neighbour | 0.4484 | 0.0173 | 0.1127 |
| Linear | 0.8881 | 0.0035 | 0.0462 |
| Random Forest | 0.9746 | 0.0008 | 0.0223 |
| Ridge | 0.4833 | 0.01487 | 0.1054 |
| Stochastic Gradient Descent | 0.8731 | 0.0046 | 0.0535 |
| Support Vector Machines | 0.8844 | 0.0045 | 0.0523 |
| Multi-layer Perceptron | 0.9303 | 0.0025 | 0.0392 |
| Extreme Gradient Boosting | 0.9963 | 0.0001 | 0.0057 |

Table 14 shows optimized and cross-validated results for each algorithm. Several similarities and differences are visible and will be explained in the following part of this research. In this part of the research, Tree-based algorithms still dominate in all evaluation values. XGboost still has the most ideal $R^2$ metrics of 0.9963, $\overline{MSE}$ of 0.0001, and $\overline{MAPE}$ of 0.0057. Furthermore, ETR, RFR, and MLP have $R^2$ values above 0.9 and the smallest evaluation errors of $\overline{MSE}$ and $\overline{MAPE}$. Other algorithms have minimal improvement or have a deterioration of metrics, and the SGD demonstrated the largest increase in $R^2$, with an increase 0.96 compared to the default hyperparameters, while LR had the smallest shift in results. The largest decreases in $\overline{MSE}$ and $\overline{MAPE}$ errors for the MLP algorithm were stark, while k-NN's performance was worst. Furthermore, as mentioned in the previous section based on the given results from Tables 13 and 14, three AI algorithms were selected for the final estimation of the excitation current in SM, namely: ETR, RFR, and XGBoost. The obtained ratios of $\sigma$ for $R^2$, $\overline{MSE}$ and $\overline{MAPE}$ are shown in the figures below.

Figure 12 shows the $R^2$ and $\sigma$ for ETR, RFR, and XGBoost algorithms. As shown in Table 14, XGboost has the highest $R^2$, followed by ETR and finally RFR; however, in this case, the main focus is on the $\sigma$ of all three algorithms, so the best result is achieved by XGBoost, which has the smallest $\sigma$ value out of all three algorithms of 0.0011, which is up to eight times lower than for ETR and RFR.

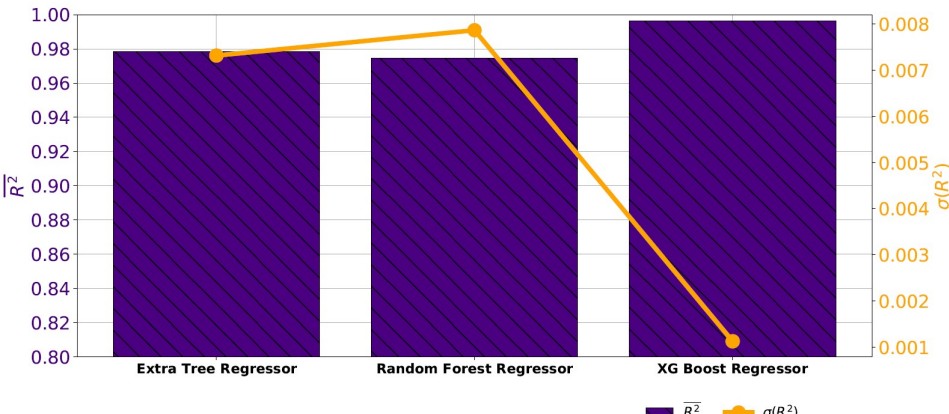

**Figure 12.** The $\sigma$ of $R^2$ results for ETR, RFR and XGBoost algorithms.

Figure 13 represents $\overline{MSE}$ and $\sigma$ for the ETR, RFR, and XGboost algorithms. It can be seen from the figure that the XGBoost again has the smallest $\overline{MSE}$ of 0.0001, while the largest in this part of the research is ETR, valued at 0.000035. The $\sigma$ for XGBoost again

achieves the best results, with a $\sigma$ result of 0.0000033, while the ETR again had the worst performance.

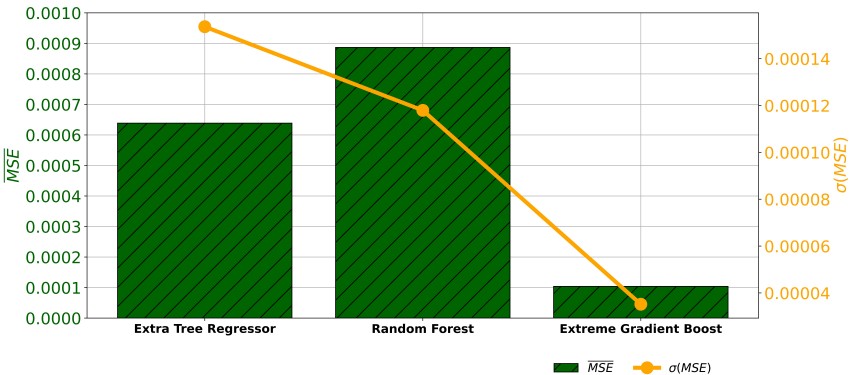

**Figure 13.** The $\sigma$ of MSE results for ETR, RFR, and XGBoost cross-validated algorithms.

In the last part of the performance evaluation of the selected algorithms, the value of MAPE and $\sigma$ of individual algorithms is shown in Figure 14. The value of the $\overline{MAPE}$ result favors (as in the previous two cases) of the XGBoost algorithm with a value of 0.0057, while the worst performance was achieved by RFR with a value of 0.0223. $\sigma$ for $\overline{MAPE}$, in this case, was 0.0003, while for RFR it was 0.0012 and for ETR it was 0.001869.

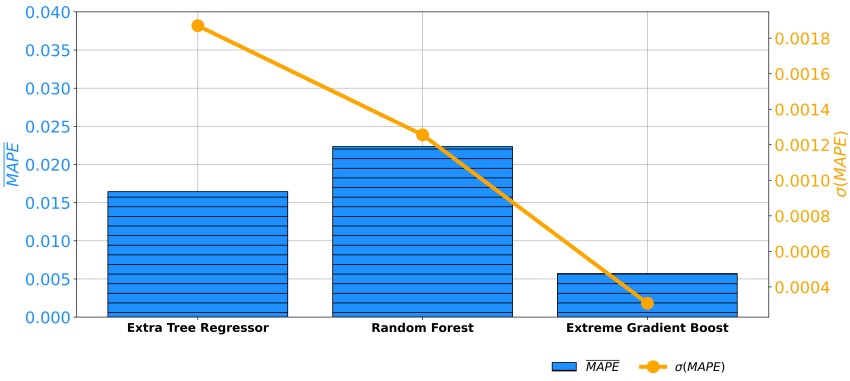

**Figure 14.** The $\sigma$ of MAPE results for ETR, RFR, and XGBoost cross-validated algorithms.

## 4. Conclusions

In this paper, different AI algorithms were used to estimate the excitation current of SM. During the investigation, three different approaches were taken to obtain a solution. The first approach was to use linear, ensemble, tree-based, and neural network algorithms to obtain favorable results for further consideration. At the same time, the hyperparameters that were taken for training the defined algorithms were set to default values. It can be seen from the results that a larger number of algorithms were considered, which shows the possibility of optimal selection for the estimation of the excitation current, but the question arose of whether the obtained model led to overfitting, i.e., whether this was the case for the affected samples in the train/test ratio. After that, 5 k-fold cross-validation of the data with randomized hyperparameters was performed. The results varied from algorithm to algorithm, with the XGBoost algorithm excelling with the best result in all three metrics. As the last part of this research, an investigation of the $\sigma$ was carried out for all three metrics; however, in this case, only the tree-based AI algorithms were considered (ETR, RFR, and XGBoost). When looking at the results for all three metrics, XGBoost again demonstrated the best results for all validation metric areas assessed. Compared to other control techniques in the literature, this model provides the following benefits: simpler and more precise results because it takes into account only the highlighted weight parameters

of the synchronous motor, and the possibility to determine the relationship between the synchronous motor parameters and the excitation current. Finally, it provides a novel approach to excitation current estimation using a different type of algorithm compared to the related research for categories of smaller motors (characteristics shown in Table 1).

Based on the conducted research, the answers to the hypothetical questions in the Introduction are as follows:

- It is possible to estimate the excitation current of a synchronous motor using an AI algorithm with high precision and accuracy. Based on the given research it was shown that the most optimal algorithm was XGBoost.
- Using GS and cross-validation, the values were validated, and the parameters of the AI model were optimized, which provides suitable evaluation metrics for the estimation of the excitation current.
- From the larger number of presented algorithms in this paper, the best possible algorithm that provides optimal results and the smallest $\sigma$ is XGboost with a high value of $R^2$ and small values of MSE and MAPE.

To conclude, and in terms of future research plans, the following points should be made. First, the given dataset should be further expanded with more up-to-date points, for example for millisecond or microsecond ratios, i.e., changes in these ratios. As a second step, the influence of heat on SM should be incorporated, as it is well known that every electrical machine, device, etc., under the influence of heat experiences changes in the given parameters. For example, a synchronous motor in low temperatures has a different resistance in relation to electrical conductors on the stator or the rotor (depending on the version). The endpoint of additional research would be to implement a given model to control logic so as to evaluate the obtained results in a real experiment. When all the mentioned points are implemented, a dataset will be created and the optimal AI model that approximates the excitation current will be obtained again; a more precise and realistic solution will be obtained concerning the research carried out in this article.

**Author Contributions:** Conceptualization, N.A., I.L. and Z.C.; methodology, N.A., M.G. and Z.C.; software, N.A., I.L. and Z.C.; validation, I.L., M.G. and Z.C.; formal analysis, N.A., M.G. and I.L.; investigation, N.A. and I.L.; resources, Z.C.; data curation, I.L. and M.G.; writing—original draft preparation, N.A., I.L. and Z.C.; writing—review and editing: N.A., M.G. and I.L.; visualization, I.L. and Z.C.; supervision, Z.C.; project administration, Z.C.; funding acquisition, Z.C. All authors have read and agreed to the published version of the manuscript.

**Funding:** This research received no external funding.

**Institutional Review Board Statement:** Not applicable.

**Informed Consent Statement:** Not applicable.

**Data Availability Statement:** The dataset is available at: https://www.kaggle.com/datasets/fedeso riano/synchronous-machine-dataset (accessed on 14 November 2022).

**Acknowledgments:** This research received support from the CEEPUS network CIII-HR-0108, European Regional Development Fund under the grant KK.01.1.1.01.0009 (DATACROSS), project, CEKOM under the grant KK.01.2.2.03.0004, Erasmus+ project WICT under the grant 2021-1-HR01-KA220-HED-000031177, and the University of Rijeka scientific grant uniri-tehnic-18-275-1447.

**Conflicts of Interest:** The authors declare no conflict of interest.

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
