# Peer review of "Estimation of Excitation Current of a Synchronous Machine Using Machine Learning Methods"

_computers, doi:10.3390/computers12010001_

Round 1

Reviewer 1 Report (Previous Reviewer 1)

This paper presents an approach to the estimation of the excitation current of a synchronous generator using machine learning methods.

The authors have responded to most of the reviewer’s previous suggestions and remarks. However, one suggestion has not been correctly addressed, namely:

-        Page 8: in line 231 you refer to “Von Mises”, while in line 235, you refer to “Von Misles”. Use the correct denomination!

Author Response

Reviewer 1

Respected Reviewer,

thank you very much for your review of our manuscript. We have tried our best to respond to the issues you have noted in your manuscript. Please find our responses below. Changes in the manuscript made due to your comments have been marked red.

Question: This paper presents an approach to the estimation of the excitation current of a synchronous generator using machine learning methods.

The authors have responded to most of the reviewer’s previous suggestions and remarks. However, one suggestion has not been correctly addressed, namely:

-        Page 8: in line 231 you refer to “Von Mises”, while in line 235, you refer to “Von Misles”. Use the correct denomination!

The answer:

Denomination of “Von Misles” is corrected to “Von Mises” in lines 328 and 332 of the new manuscript version, therefore authors do apologize for mistake made while writing. We hope you will be satisfied changed version of manuscript and correct denomination.

Kindest regards,
The Authors

Reviewer 2 Report (Previous Reviewer 2)

There are no comments.

Author Response

Respected Reviewerm

thank you for your time and effort for reviewing our manuscript.

Kindest regards

The Authors

Reviewer 3 Report (Previous Reviewer 3)

In my opinion, 

after the maked changes and prepared answers for all my questions, the articel is much better, interesting and underestanding for readers.

The authors: 

- noticed errors when re-examining the paper,

- applied these corrections to the entire work,

- in this way they have contributed to the quality of the written work,

- emphasized in the paper the contribution of this articel in the Introduction section and in the Conclusion section, 

- carried out measurement verification and it was added and described in subsection 2.2 in the initial research by the original authors of the dataset. 

The reviewer

Author Response

Respected Reviewer,

thank you very much for the detailed analysis of our manuscript, and thank you for your insight into improving the quality of the article in a technical and scientific sense.

Kindest regards,

The Authors

Reviewer 4 Report (Previous Reviewer 4)

The effort devoted by the authors to improve their manuscript according to the recommendations of the first three distinguished reviewers is to be acknowledged. Unfortunately, in what concerns my remarks, I consider them not properly addressed. In my opinion, with all the sincere appreciation for the hard work involved, the approach is basically flawed, being impossible to bypass/ignore the electrical machine’s equations, which carry in fact the physical validity and consistency of electromagnetics. The aforementioned device is a 100 percent deterministic electromagnetic system, and it should be treated as such. This assertion doesn’t mean that any modern AI tools couldn’t be imagined and applied to improve its design and performance: weight, efficiency, losses, torque, better materials to be used, etc., but all in accordance with the machine’s equations. In particular, that applies to the excitation current (one of the dozens of functioning and design parameters of the machine). AI can be utilized for tolerances of the parts, errors in establishing the right voltage/current, etc., but all have behind the machine’s equations. The presented methods could be valuable per se or in the case of other non-deterministic systems or exhibit fuzzy mathematical connections between parameters: sociology, economics, medicine, meteorology, oceanography, etc.

Author Response

Response to reviewer 4

The authors of this manuscript want to thank the reviewer for his time and effort to give constructive comments and suggestions which improved the quality of the manuscript. The authors of the manuscript hope that manuscript in this form is acceptable for publication. All changes in the manuscript are highlighted in blue. The answer to the comments provided by the reviewer are given below.

The effort devoted by the authors to improve their manuscript according to the recommendations of the first three distinguished reviewers is to be acknowledged. Unfortunately, in what concerns my remarks, I consider them not properly addressed. In my opinion, with all the sincere appreciation for the hard work involved, the approach is flawed, being impossible to bypass/ignore the electrical machine’s equations, which carry the physical validity and consistency of electromagnetics. The aforementioned device is a 100 percent deterministic electromagnetic system and should be treated as such.

Answer: Authors want to draw the attention of the reviewer to the basics of mathematical modeling. Any real system that is being mathematically modeled is a non-linear example, a simple mathematical pendulum is a non-linear system i.e. described by non-a linear differential equation if we include various effects such as air drag that occurs during the movement of the pendulum. The system can be described with a simple differential equation by neglecting the additional influences on the system in this case air drag.

Regarding the synchronous machine system analyzed in this paper, the system can be described with simple mathematical equations if we neglect a bunch of other parameters that could influence excitation current. Most likely, this is how it is done in practice and that is why the reviewer does not see the manuscript's scientific contribution. However, in scientific research of such a system in order to mathematically model the system as close as possible to the real system some number of non-linear effects have to be included (more the better). For example, due to the non-linear characteristic of magnetization, the magnetic flux in the saturation region, salient poles have a large number of coupled variables and high nonlinearity voltage and current relationships in the machine that are not linear, nonlinear relationship between flux and mmf, etc.

If these effects are included in the mathematical model the system analysis and calculation become complex and it is hard to solve using conventional methods.

The authors also want to draw attention to a large number of published papers in which other authors stated that synchronous machines are complex systems meaning that these systems are non-linear. If this is not true then the reviewer should explain to us how is it possible to achieve the estimation of excitation current with 100% accuracy in any case possible using so-called “electrical equations”. We would be very interested in this explanation.

If this is true then any published paper, book or other types of in which artificial intelligence was applied does not have any sense and should be revoked. Several examples for examination:

  • Bayindir, Ramazan, et al. "Application of adaptive artificial neural network method to model the excitation currents of synchronous motors." 2012 11th International Conference on Machine Learning and Applications. Vol. 2. IEEE, 2012.,
  • Kahraman, Hamdi Tolga. "Metaheuristic linear modeling technique for estimating the excitation current of a synchronous motor." Turkish Journal of Electrical Engineering and Computer Sciences6 (2014): 1637-1652.,
  • Kaplan, Orhan, and Emre Elik. "Simplified model and genetic algorithm based simulated annealing approach for excitation current estimation of synchronous motor." Advances in electrical and computer engineering4 (2018): 75-84.,
  • Lu, Wenzhe, Ali Keyhani, and Abbas Fardoun. "Neural network-based modeling and parameter identification of switched reluctance motors." IEEE transactions on energy conversion2 (2003): 284-290.,
  • Vas, Peter. Artificial-intelligence-based electrical machines and drives: application of fuzzy, neural, fuzzy-neural, and genetic-algorithm-based techniques. Vol. 45. Oxford university press, 1999.,
  • Del Angel, Alberto, et al. "Estimation of rotor angles of synchronous machines using artificial neural networks and local PMU-based quantities." Neurocomputing16-18 (2007): 2668-2678.,
  • Parvizi, Amin. "Artificial Intelligence Techniques of Estimating of Torque for 8: 6 Switched Reluctance Motor." Fuzzy Logic: Algorithms, Techniques and Implementations(2012): 193.,
  • Toh, Allan KP. Artificial neural network flux estimation for field oriented control. University of Calgary, 1994.,
  • Pillutla, S., A. Keyhani, and I. Kamwa. "Neural network observers for on-line tracking of synchronous generator parameters." IEEE Transactions on Energy Conversion1 (1999): 23-30.,
  • Ramana, Pilla, Karlapudy Alice Mary, and Munagala Surya Kalavathi. "A novel parameter estimation method for permanent magnet synchronous motor drive." International Journal of Power and Energy Conversion3 (2018): 295-310. ,
  • Sabanci, Kadir. "Artificial intelligence based power consumption estimation of two-phase brushless DC motor according to FEA parametric simulation." Measurement155 (2020): 107553. and
  •  

The publicly available dataset that was used in this research consists of 4 input variables Load current, power factor, power factor error, changing of excitation current of synchronous machine, and excitation current as the target variable. These input and output variables were measured in the real system. First of all, these variables are non-linear (caused by nonlinearity inside other parameters in the machine, and performed measurements) which means they exhibit non-linear behavior. Severity influences are on the parameters given in the dataset such as nonlinearity of magnetic flux caused by B-H characteristics of sheet packages in the machine, inducing a voltage at a constant rotor speed i.e. non-linearity dependence of the induced voltage on the magnetic torque of the rotor caused by saturation, non-linear of character from a given flow or induction at a certain place into the machine (most often in air gap). But in this specific example, all of the parameters are non-linear, there is not a single parameter in the dataset that has a constant linear characteristic. Parameters in the dataset are visible in Figure 1.

Figure 1. Representation of parameter behavior in the dataset measurements.

Secondly, the measurement equipment which was used in this experimental investigation also has some errors in measuring. As the reviewer probably knows any sensor has some deviation from the real value in an ideal environment (perfectly sterile, temperature, humidity, and pressure consisted environment). This error can increase due to measurements conducted in an unideal environment.

The problem that was being investigated in this paper was not the dataset and obtaining these values with the smallest error possible. The idea of this paper was to obtain the highest estimation accuracy possible with the publicly available dataset.

The authors agree with the reviewer that electromagnetism in general is a linear theory since Maxwell’s equations are linear. (https://physics.stackexchange.com/questions/362705/why-is-classical-electromagnetism-linear), however, the entire system (synchronous machine) is non-linear and complex. To emphasize these four input variables are non-linear, the measured values certainly slightly deviate from real ones due to the equipment used to measure these parameters. 

This assertion doesn’t mean that any modern AI tools couldn’t be imagined and applied to improve its design and performance: weight, efficiency, losses, torque, better materials to be used, etc., but all in accordance with the machine’s equations. In particular, that applies to the excitation current (one of the dozens of functioning and design parameters of the machine). AI can be utilized for tolerances of the parts, errors in establishing the right voltage/current, etc., but all have behind the machine’s equations. The presented methods could be valuable per se or in the case of other non-deterministic systems or exhibit fuzzy mathematical connections between parameters: sociology, economics, medicine, meteorology, oceanography, etc.

Again the idea of this paper was to utilize different machine learning methods to estimate the excitation current with high accuracy. Since AI methods learn from the provided dataset the idea was to investigate if using only those four input variables the excitation current could be estimated and if so if it could estimate with high estimation accuracy. As seen in the investigation all AI methods learned from the given data and could estimate the excitation current with high accuracy.

To summarize the comments addressed to the reviewer if this is a linear problem in which using simple equations all the parameters in the dataset could be calculated with 100% accuracy then all the papers regarding the investigation using different scientific approaches even papers regarding estimation using AI methods should be revoked since they have no sense at all.

And finally, to improve the quality of the article, more literature was added that indicates the complexity of nonlinearity and the very importance of applying artificial intelligence in electrical machines. Marked with supposed signs in the Introduction section with blue color.

AC electrical machine loads used in the industry draw reactive power from the electrical power grid. Reactive power is a disadvantage, i.e. a problem, for the reason that it causes overloading effects on the power grid, switches, transformers, and relays, and unfortunately, reactive power cannot be converted into useful, i.e., mechanical energy. To normalize the amount of reactive power in the power grid, SMs are used as reactive power compensators [23]. To obtain high-quality regulation and compensation of reactive power, it is necessary to regulate the SMs power factor parameter which can be done with proper regulation of the excitation current. The value of the excitation current dictates the operation mode of the synchronous motor (capacitive or inductive character) and also affects the stability of the system [24].

The main problem is that most EM manufacturers do not provide enough information about the machine, which reduces the possibility of achieving highly efficient control. In general, manufacturers provide information on rated output power, rated maximum speed, rated input voltage, rated current, protection level, dimensions, and weight. However, a minority of manufacturers provide more important information, for example, the speed-torque curve (most often at the customer's request). Often, the nominal parameters of the synchronous motor are available and sufficient for designing the regulation, but the problem is the non-linearity of the parameters, which is noticeable in an unadjusted operating environment (for example, high or low ambient temperature) or not adapted to operating conditions (for example, speed or load torque) [25,26].

Apart from the non-linearity, external and internal conditions affecting EM, the main problem is that there is no clear relationship between the parameters of the synchronous motor [27-30] The parameters of SM are mostly complex and non-linear thus, modeling SM parameters such as excitation current, power factor, and load current when the synchronous motor is running on lagging, leading or unity condition for reactive power compensation, is a difficult task [24,27] With the aim of better quality improvement of SM modeling and more precise estimation of parameters, many scientists undergo artificial intelligence (AI) estimation methods. Estimation of model parameters using techniques for linear systems became after many years perfected and frequent. However, more and more physical systems indicate non-linearities when increasing the dynamic range of high-performance equipment. Non-linearities in the real world are mostly ignored, but under the assumption that linear system theories are applicable for retrieval of less accurate approximate solutions. For industrial applications, these solutions are within acceptable limits, but for applications in high-efficiency machines, these types of linear systems are inadequate. With this principle, there is a need for further development of advanced model identification solutions for non-linear systems such as the synchronous motor. After defining the obstacles that are typical for the highly coupled nonlinear system such as synchronous motors, the authors used Particle Swarm Optimization (PSO) in [27] to obtain a high-quality model with a low error rate that is robust and generally applicable to other similar systems with a nonlinear structure. Various optimization algorithms are used to optimize permanent magnet SM (PMSM) such as evolutionary algorithm, ANN, artificial bee colony (ABC), immune method (IM), whale optimization method (WOM), particle swarm optimization (PSO) method, flower pollination (FP) method, cuttlefish optimization algorithm, and genetic algorithm (GA) that are in detail analyzed in the state of the art by [25,31]. The parameters of the PMSM model tend to change, i.e. the influence of non-linearity of parameters due to temperature and aging online techniques are used to identify more up-to-date parameters to design a more robust control [32].

In addition to potential optimization, AI is used to obtain improved waveforms at the output and reduce oscillations (such as output speed, torque, and current variations for three phases). For example, using a fuzzy logic controller in combination with AI with certain conditions (in this case with 25 conditions and 49 conditions) results in a much smoother rotation of the engine and thus less oscillation, which greatly contributes to the improvement of the system [33].

Using AI more precisely, Fuzzy logic in combination with ANN represents an advanced method that is applied for AM control logic. AM is a rather non-linear machine where the influence of temperature, age, and additional vibration elements relations in electromagnetism, affects the operation of the machine, so with this idea, the authors, after previously mathematically modeling AM, defined a control strategy based on rotor flux in the work [34], which gives the robustness of the algorithm.

Regarding synchronous motors with electromagnets, in the article [35]  the authors indicate the complexity and nonlinearity of the SM parameters. By applying the symbiotic organisms search (SOS) algorithm, gravitational search algorithm (GSA), ABC, and GA, authors investigate the possibility of obtaining a high-quality algorithm with a small error of approximation, whereby the best results are achieved by SOS with a maximum error of 0.1703A. “

Additionally to describe the imperfections of synchronous machine and the possibility of exact values while calculating the parameters are describe in subsubsection 2.0.1 “Potential challenges when modeling a synchronous motor” are as follows and marked with quotation marks:

In modern EM modeling, the contribution of the author G.Kron [44,45] is important, where the general theoretical background for EM modeling is set, but most often neglected. The generality of Kron's tensor-based approach is diminished while using matrices. By using matrices, EMs are mostly treated as magnetic (deterministic) linear systems, while nonlinear properties are ignored. However, to create positive ratios between measured and calculated values and to be applied in nonlinear control modeling, it must be included in modeling SM.

In addition to neglected factors when modeling a synchronous machine, there are several losses in SM that make the machine complex and nondeterministic and require additional power (iron losses, winding losses, and ventilation losses)[46]. Of course, these losses occur depending on the situation in which the machines are but it makes it unpredictable to estimate parameters with regular measuring methods. Certain calculations can be done to gather information, but due to the complexity of thermal effects in EM, numerical calculation methods such as electromagnetic and thermal Finite Element Modeling (FEM) or Computational Fluid Dynamics (CFD) are only used for a specific analysis of EM part (part of the stator core for example): A a complete thermal analysis would require a full CFD/FEM coupling.

As for the modeling of the SM, the turn insulation in the field winding in the case of the silent pole synchronous machine (generator) is generally thin, so it is often neglected in calculations. However, surface insulation is of similar dimensions, almost equal to the surface of the conductor, so it would affect the result of the heat calculation. On the other hand, in the calculation of the temperature field of a synchronous machine with silent pole rotor insulation, i.e. the insulating layer, is often neglected, while even at the smallest insulation there is a temperature drop, which can certainly affect the modeling of the EM itself [47].

This subsection indicates the imperfection of calculations when modeling or estimating EM. This raises the importance of using AI algorithms for the optimization or estimation of synchronous machine parameters. AI algorithms are based on the data contained in the dataset, and the more data the dataset (algorithm) has, the greater the robustness and readiness, as well as the accuracy of the obtained model is. In the continuation of this research, the parameters of the used synchronous motor and the statistical analysis and distribution of data for input into the AI model are described.

Round 2

Reviewer 4 Report (Previous Reviewer 4)

I maintain my strong belief that the assessment and a better characterization of an electromagnetic device (i.e., the synchronous machine in the present case) cannot avoid electromagnetism at its basic foundation. Thus, any AI/stochastic approach should incorporate in one way or another the laws of electromagnetism, represented either by the classical equations of the machine or any other relationship (possibly improved) derived from these laws. Any linear model can be adjusted to account for any nonlinearity and/or any other physically relevant phenomenon having a significant role to play in the overall system of the machine. I disagree with the point of view of the authors that nonlinearities, B-H relationship for example, and some other factors (even numerous, possibly) that influence the functioning of the machine, all represent as many justifications that electromagnetism, mechanics, etc. should be dropped, entirely. Modern multiphysics numerical simulation software all incorporate nonlinear materials and phenomena, relying entirely on the laws of physics as their name suggests (mechanics, electromagnetics, thermotechnics, material science, etc.). Specific to the present problem is that the deterministic part overwhelmingly outweighs the non-deterministic part (fabrication tolerances, environmental parameters, etc.) Therefore, a robust approach (incorporating any modern AI algorithms) should rely on deterministic physical equations. The alternative proposed by the authors should be to rely exclusively on sets of measurements performed on the machine itself, followed by AI processing. That would imply that for each and every machine on the market, a laboratory and its personnel should be available to perform these extensive sets of measurements, to be later processed by AI algorithms. Is this approach technically and economically feasible, in the first place? What would be then the role of design and simulation in engineering? To conclude, in my opinion, the presented AI procedures are interesting and valuable per se, but they simply do not apply to mostly deterministic systems like the one under scrutiny in the proposed manuscript.

This manuscript is a resubmission of an earlier submission. The following is a list of the peer review reports and author responses from that submission.

Round 1

Reviewer 1 Report

This paper presents an approach to the estimation of the excitation current of a synchronous generator using machine learning methods.

The authors should take into consideration the following issues:

-        Page 1, line 23: sentence “… constructed by his purpose” must be rephrased!

-        Page 3: at the end of the introductory section, the authors should clearly state and present the original contributions of the proposed paper.

-        Page 6: pay attention to repeated sentences – sentence in lines 179-180 is a repetition of sentence in lines 174-175.

-        Page 6, lines 182-183: explain what do you understand by “values at 25%, 50%, and 75% of the maximum values”.

-        Page 7: in line 199 you refer to “von Mises”, while in line 203, you refer to “Von Misles”. Use the correct denomination!

Reviewer 2 Report

1. The authors state in the text that “the only difference between AM and SM is the speed of rotation of the rotor”. It is known that there are several other differences, among which the aspects of design, construction, drive, in short, can be mentioned. In this sense, authors should consider other aspects in the text of lines 151 and 152 of the proposed article.

2. Figure 5 shows the relationship between the load angle (δ) and the torque T produced. Thus, it is important to graphically characterize the mentioned load angle due to its importance, in order to make this aspect more explicit to the reader.

3. In dataset analysis it is mentioned that the data consists of 4 input parameters. Comment why 5 input parameters are shown in Table 1.

4. Inform what are the parameters Io in equation (3) and φ in equation (5).

5. What is σ used in item 3 of the article?

6. Finally, it is necessary to cite Figure 11 in the text (line 438).

Reviewer 3 Report

Dear Authors,

thank You for the possibility to read and review Your work.

Dear Authors,

thank You for the possibility to read and review Your work: “Estimation of excitation current of a synchronous machine using machine learning methods.

First, I would like to note that the article could be improved in terms of editing and checked for linguistic correctness of specialized phrases in technical English.

The readers should have clearly given the impact of described methods, madding changes and the effectiveness of the methods should be more emphasized.

I. My general questions and comments are as follows:

1. What type of machines are you researching?

2. What are its rated parameters? Is it generator or motor operation? Silent-pole or round rotor type pole rotor or doesn't it matter?

3. Turbogenerator or hydrogenator was examined?

4. The article provides an overview of data sheets in terms of several parameters - if these data sheets are for a series of types, please specify for which. What voltage is it supplied with and what power?

5. Why is it so important to estimate only the excitation current? Is it big power synchronous machine? What about the other parameters?

6. In my opinion, the synchronous machine has been too general presented and too much simplified.

7. Have any experimental studies been performed prior to modeling and simulation?

8. Do you have the no-load and steady short-circuit characteristics of a synchronous machine?

9. Do you have verification measurements and they have been carried out?

10. What does the test stand look like? Please provide a block or electrical diagram.

11. Was the stability of the equivalent parameters of the synchronous machine model in the d and q axes assumed to be constant?

12. Has the armature effect been disregarded for consideration?

13. Has the machine stator resistance been considered or no influence?

14. Has the dependence of the Xd synchronous reactance on the excitation current value been determined?

15. How is the load of different nature taken into account for the operation of a synchronous generator and the need to know the voltage changes at the terminals due to changes in the load current?

16. Assuming the linearity of the magnetic circuit, the magnitudes of the voltage and the current of the catheter are linearly dependent, so the characteristic is a straight line - the voltage at the terminals increases linearly with the increase of the current value, and assuming the linearity of the magnetic circuit, it is not possible to obtain the short-circuit point. It is obvious that due to the saturation of the magnetic circuit, the voltage will increase according to a non-linear relationship. The change in voltage at the terminals, with a constant value of the electromotive force, decreases with increasing current in a non-linear manner. Do you take into account the saturation?

17. It is important to know the dependence of the voltage at the terminals as a function of the load current when using a generator to ensure power supply in the event of a power outage. Do you know or have it?

18. The dependence of the excitation current on the armature current, at a constant voltage value at the terminals and a constant rotational speed (control characteristic), has a shape depending on the nature of the load. The actual shape of the characteristics is the result of changes in the values of inductance, resistance or capacitance. In this case, for example with a capacitive load, the maximum value of the current occurs at a capacitive impedance equal to the synchronous reactance (voltage resonance). Do you take into account it?

19. Have you measured or the V curves of a synchronous generator been determined?

20. In the case of high power generators, it is not possible to experimentally determine all the nominal values of a synchronous machine. In order to estimate the value of the rated excitation current and voltage variation, it is necessary to determine the leakage reactance and estimate the magnetic field produced by the rated armature current (armature response) on the magnetizing current scale. This is the case for high power generators only.

21. Has the Potier Triangle been determined, which is necessary for the determination of the rated excitation current and voltage variation of a synchronous generator?

22. Most synchronous generators operating in the power grid should be operated with a symmetrical load. Has the influence of the component of the magnetic field rotating against the direction of rotation been taken into account? The rotor shaft, which induces tensions in the starting-damping cage or in the solid material of the core of a fully polar machine?

II. My questions and comments by lines:

Was the title of the article properly defined? Title suggest the possibility to estimate the actual value of excitation current in SM for different states but it is not exactly true in this article? SM is nonlinear object with a lot of dependencies - please describe the application range and filed, check and attach the Mordey V characteristic before and after estimation. For what we could use the estimation in practice?

Line 6 - The purpose of this paper is to estimate the excitation current on a publicly available dataset, from the following input parameters Iy: Load Current, PF: Power factor, ePF: Power factor error, dif: - please check the Mordey V characteristics.

Line 6- Changing of excitation current of synchronous machine, by using artificial intelligence algorithms. – How high power machines was taken? How does look like the AI algorithms?

Line 121 – “Considering the high accuracy rate of AI algorithms in the area of prediction and regression, the following hypothesis questions are raised regarding SM: is it possible to estimate precision rate and a small evaluation error, is it possible to optimize the model and confirm the obtained results with 5 k-fold, cross-validation using the randomized hyperparameter search and which algorithm gives the best results with the possibility of implementation in a  real-life situation.” – How we could check it? For what kind of SM?

Line 181 – Table 1 – What kind of types data sheet was analysed? What kind of machines? How high power?

Line 456 - In this paper, different AI algorithms were used to estimate the excitation current of SM – the data sheet nominal current is it only one point in nonlinear parameters machine. How we could realy check this?

Line 460 - It can be seen from the results that a larger number of algorithms were considered, which shows the possibility optimal for selection for the estimation of excitation current, but the question arose whether the obtained model was overfitting, i.e. whether the affected samples during the train/test ratio went in that direction. – What kind of optimalization method? What kind of criteria?

Line 470 - Based on the conducted research and given hypothetical questions in the Introduction in the 1 section, the answers are as follows:

• that it is possible to estimate the excitation current of SM using an AI algorithm with high precision and accuracy, based on the given research it was shown that the most optimal algorithm was XGBoost, - If we take other nominal parameters we could receive better or worse effects?

• using GS and cross-validation, the values were validated and the parameters of the AI model were optimized, which provides suitable evaluation metrics for the estimation of excitation current and – how was the optimalization conducted?

• from the larger number of available algorithms and ten tested algorithms presented in this paper, the best possible algorithm that gives the most optimal results and the smallest σ are XGboost with a high amount of R2 and small amounts of MSE and MAPE. – please consider supplementing the article with a flow chart for selected methods and classification mechanism, then applicability and use of the proposed methods?

Line 484 - First, the given dataset should be further expanded with more up-to-date points, for example in the ratio of milliseconds or microseconds, i.e. changes in these ratios – please give more details?

Line 484 - As a second step, the influence of heat on SM should be added, as it is well known that every electrical machine, device, etc. under the influence of heat has changed to the given parameters. – My base question – for what kind of SM type date was choose and analysed? For what kind of conditions and real rules? What was implemented and was simplified or not take into account?

Line 487 - For example, SM in low temperatures has a different resistance of electrical conductors on the stator or the rotor (depending on the version). – Yes, but first are very important the working states of SM – I asked before in point I.

Line 489 - The endpoint of additional research would be to create a set of data with several points in real conditions, that is, to connect a variable load to the SM shaft, and use a combination of these parameters to measure the values of individual variables of the synchronous machine. – So, as I wrote, please define the conditions for the article, the working states and analysed parameters for better understanding.

Line 491 - When all the mentioned points are implemented, a dataset will be created and the optimal AI model that approximates the excitation current will be obtained again, a more precise and realistic solution will be obtained concerning the research carried out in this article. – please make this on the received data?

III. My questions and comments:

1. What is the practical purpose of the described methods in industrial applications or systems?

2. What we receive thanks to proposed methods?

3. What types of SM motors in terms of their parameters (power, currents, torque, design) can be analyzed using the methods?

4. In what scope of construction, power, type of construction, diameter, will your methods be effectives?

5. How influence on the accuracy of the method (relative errors) the influence of the magnet technology and the precision of serial production, as well as the motor construction and construction of magnets for SM motors produced in series?

6. What we could expected when applied the proposed SM estimation of excitation current relate to cooperation in servo controller for PMSM and an IGBT inverter?

7. What is the impact of the simplification assumptions on the accuracy of proposed methods? Higher harmonics, passive moments and unfavorable accompanying phenomena are generated for industrial systems.

8. How the proposed methods finally performs the analysis to calculate the irregular magnetic poles with arbitrary section shape in principle?

9. What conditions must be met technologically in real SM to ensure the universality of this methods?

10. Is it possible to present graphically how are the data SM changes for the calculated parameters and assuming the operating conditions?

I hope that my comments will help to improve the article.

Kind regards

Reviewer

Reviewer 4 Report

The proposed manuscript is devoted to predicting the synchronous machine (SM) excitation current using several artificial intelligence (AI) procedures. The introductory part containing the description of the SM enters into too many details, generally known by professionals in the field. This presentation resembles undergraduate lecture notes, without bringing any novelty to the reader or without further relevance for the considered AI approach (the original part of the paper). In contrast to this oversize part, the actual problem statement of the proposed research is largely overlooked. What is the goal of the research and which are its initial parameters, all represent insufficiently discussed aspects. It is not stated which is the considered operation regime (motor/generator) of the SM, its rated parameters, the hypothesis and the context arising from the necessity of excitation current prediction, the mechanical load characteristic-if the SM motor is investigated, or the consumer impedance-if the SM generator is considered, etc. To conclude, the reader is presented with an array of regression methods for determining the SM excitation current and several associated parameters without knowing the motivation of the proposed undertaking and also some essential initial data that normally should have been disclosed from the very beginning. This lack of information gap is further enlarged by the scarcity of the description concerning the used regression methods principles. The authors present information regarding the fine-tuning of the methods (via some specialized parameters) without a brief presentation of the method itself, and how these specialized parameters come into play to achieve the final goal of the research. Another concern is whether or not the specific equations of the synchronous machine are used at some point in the research. How to predict the excitation current without any reliable and realistic model of the device and its associated equations? To put it in other words, is it possible for the SM to replace the classical approach based on electromagnetism (be it circuit theory and/or field theory) with probabilistic approaches combined with AI regression methods? On which grounds one can guarantee that this approach is valid? Is there some proof?

 Last but not least, the choice for the Energies journal is less evident. Indeed, electrical machines are energy or power converters. But here, in the proposed manuscript, the emphasis falls on totally different technical aspects, namely AI, regression methods, etc.